# The Effects of COVID-19 on Skeletal Muscles, Muscle Fatigue and Rehabilitation Programs Outcomes

**DOI:** 10.3390/medicina58091199

**Published:** 2022-09-01

**Authors:** Camelia Corina Pescaru, Adelina Marițescu, Emanuela Oana Costin, Daniel Trăilă, Monica Steluța Marc, Ana Adriana Trușculescu, Andrei Pescaru, Cristian Iulian Oancea

**Affiliations:** 1Pulmonology Department, ‘Victor Babes’ University of Medicine and Pharmacy, Eftimie Murgu Square 2, 300041 Timișoara, Romania; 2Center for Research and Innovation in Precision Medicine of Respiratory Diseases (CRIPMRD), ‘Victor Babes’ University of Medicine and Pharmacy, 300041 Timișoara, Romania; 3‘Victor Babes’ University of Medicine and Pharmacy, Eftimie Murgu Square 2, 300041 Timișoara, Romania

**Keywords:** SARS-CoV-2, COVID-19, muscle weakness, muscle disease, muscle fatigue

## Abstract

*Background and Objectives*: Consequences due to infection with SARS-CoV-2 virus can have a direct impact on skeletal muscle, due to the fact that both cardiac and skeletal muscle tissue show robust ACE2(angiotensin-converting enzyme 2) expression, suggesting a potential susceptibility to SARS-CoV-2 infection in both types of tissues. From the articles analyzed we concluded that the musculoskeletal damage is firstly produced by the inflammatory effects, cytokine storm and muscle catabolism. However, myopathy, polyneuropathy and therapies such as corticoids were also considered important factors in muscle fatigue and functional incapacity. Pulmonary rehabilitation programs and early mobilization had a highly contribution during the acute phase and post-illness recovery process and helped patients to reduce dyspnea, increase the capacity of physical effort, overcome psychological disorders and improved the quality of their life. *Materials and Methods*: We have included in this review 33 articles that contain data on muscle damage following SARS-CoV-2 infection. We used the following keywords to search for articles: SARS-CoV-2, COVID-19, muscle weakness, muscle disease, muscle fatigue, neurological disorders. As a search strategy we used PubMed, Cochrane Database of Systematic Reviews; Database of Abstracts of Reviews of Effects and Health Technology Assessment Database to collect the information. We also have chosen the most recent articles published in the last 5 years. *Conclusions*: Muscular damage, as well as the decrease in the quality of life, are often a consequence of severe SARS-CoV-2 infection through: systemic inflammation, corticotherapy, prolonged bed rest and other unknown factors. Pulmonary rehabilitation programs and early mobilization had a highly contribution during the acute phase and post-illness recovery process and helped patients to reduce dyspnea, increase the capacity of physical effort, overcome psychological disorders and improve the quality of their life.

## 1. Introduction

The main scene of COVID-19 infection seems to be the respiratory tract and epithelium, where type II pneumocyte expresses a large number of ACE2(angiotensin-converting enzyme 2) and TMPRSS2(transmembrane serine protease 2) receptors, being a consequence of the development of viremia [1]. The ACE2 and TMPRSS2 receptors were also highlighted in the smooth muscle tissue, endangered, fibroblasts and articular cartilage, and in the cortical and trabecular bone tissue, the presence of the ACE2 receptor was identified in large proportions [2]. These findings suggest that muscles, bone tissue and cartilage are practically hosts for the SARS-CoV-2 virus, which is why symptoms such as myalgia and muscle weakness, bone and joint pain reported in infected patients, are justified due to the inflammatory process developed in these cells [3]. This inflammatory storm is documented at the biological level by the presence of increased levels of the enzyme creatinkine and lactate dehydrogenase, justifying muscle dysfunction and motor deficiency in patients infected with the SARS-CoV-2 virus [4].

The exact biological mechanisms of this virus on skeletal muscle are not entirely understood. The studies carried out so far on rodents have shown a decrease in muscle mass, only 4 days after infection. Moreover, the autopsies of the patients who died as a result of the COVID-19 disease, brought information about the nature of the muscle disorders [5]. It was observed in these patients the presence of muscle atrophy due to a necrosis of the muscle fibers and the presence of inflammatory cell infiltrates. After making electron micrographs, there was a disorganization of myofibrils, of disk Z and a suggestive neuronal demyelination [6,7]. These findings support symptoms such as fatigue, decreased muscle strength and fatigue, weakness and physical stress, described by COVID-19 patients [7]. The manifestations of COVID-19 are predominantly respiratory, but the clinical picture is enriched by signs and symptoms with extrapulmonary involvement [8]. According to some reports and studies, COVID-19 can cause acute and long-term neurological complications in these patients [9]. Although neurological symptoms and manifestations are rare and small compared to respiratory ones, while the pandemic progresses, the proportion of patients with neurological manifestations is expected to increase [10].

## 2. Materials and Methods

We have included in this review 33 articles that contain data on muscle damage following SARS-CoV-2 infection. We used the following keywords to search for articles: SARS-CoV-2, COVID-19, muscle weakness, muscle disease, muscle fatigue, neurological disorders. As a search strategy we used PubMed, Cochrane Database of Systematic Reviews; Database of Abstracts of Reviews of Effects and Health Technology Assessment Database to collect the information. We also have chosen the most recent articles published in the last 5 years.

## 3. Results

Following the research carried out, we have summarized the result in the table and figure below (Table 1), (Figure 1).

## 4. Discussion

### 4.1. Cytokines and Skeletal Muscle

Receptors for SARS-CoV-2, ACE2 and TMPRSS2 are present in a wide variety of human cells, which is why COVID-19 targets on different systems from the human body.

The virus binds to these receptors, enters the cells, where the proteolysis of the viral S protein takes place, which releases a cellular signal. Following that, the viral compounds mix with the human ones and the viral RNA is released in the cytoplasm of the host cell [11]. Once viral RNA penetrates the cytoplasm, it can be replicated; the virions are released from the host cells by exocytosis to other cells. Cellular proteins encoded by viral RNA can produce dysfunction of other cells by interacting with those healthy human cells [41]. Among the dysfunctions produced by these phenomena, there are also the pathological respiratory phenomena occurring at the mitochondrial level, at the cyto-skeletal level or at the level of nuclear transport. These disorders of cellular function can even lead to death of the target cells. Also, the apoptosis of the cells emphasizes the inflammatory process already present in an infected organism, and this inflammatory process induces and supports the musculoskeletal fragility [42]. Therefore, patients who developed moderate to severe forms of the disease, and even those with relatively mild forms, felt an important muscle burden, as well as bone and joint disorders [6,11]. Disser et al. explained notions related to the biological mechanisms triggered by the COVID-19 infection. Due to the similarity between SARS-CoV-1 and SARS-CoV-2, studies on the mechanisms of penetration into the human body and cell destruction went much faster. Thus, it is known that the two viruses enter the human cell attaching to the ACE2 receptor (angiotensin converting enzyme 2), present at the level of the target cells. After this binding has been achieved, the viral S protein suffers a proteolysis phenomenon achieved by proatase TMPRSS2 (serum transmembrane protease 2). Viral peptide elements will thus merge with cellular elements and viral RNA will be formed and released into the cytoplasm. Subsequently, the replication of the viral RNA occurs; following that, the new formed virions will be released from the host cell by exocytosis. Proteins encoded by viral RNA end up interacting with other proteins in healthy cells, leading to disruption of their function. After several analyses of previous human sequencing data, it was identified which tissues express these proteolytic receptors and enzymes necessary for viral replication. Thus, at the level of the skeletal muscle cell, numerous proteas of the TMPRSS2 type have been highlighted, while at the level of the smooth muscle cell, ACE2 receptors have been identified. The cells in the synovial membranes have expressed both ACE2 receptors and TMPRSS2 proteases, and the chondrocytes and fibrocondrocyte seem to present numerous ACE2 receptors. Their presence explains the accelerated viral replication, the appearance of massive cellular destructures, because these muscles, cortical and synovial tissues are practically direct loci of SARS-CoV-2 infection [11].

After the comparisons made between the pandemic of 2002–2004 with the SARS-CoV-1 virus and the current pandemic with the SARS-CoV-2 virus, the following findings were reached. Patients infected with SARS-CoV-2 virus, with moderate and severe forms of the disease, reported the appearance of muscle damage, decreased exercise capacity and marked fatiguability.

### 4.2. Proinflammatory Viral Effects

It is well known that SARS-CoV-2 produces an enormous inflammatory response that comes with the damage of a multiple structures. The phenomenon of the immunological response is the cytokine storm, which is defined by an excessive liberation and activation of inflamatory cytokines and chemoattractants, substances produced by the interation of the virus with the immune system of the host. The cytokines’ reaction is characterized by a cell infiltration with mononuclear cells in multiple tissues and also lymphopenia, which causes modifications into the patient’s blood tests, due to the severity of the disease and the hyperinflammation and an increased number of neutrophil and lymphocytes due to the depletion of T cells [43,44].

T-cell depletion may occur as a reaction to the excessive cytokine signals, with the implication and disturbance of both innate and adaptive immune response. That is the explanation for why people have different sympthoms and evolutions, since they present different intensities of immune responses, based on each profile and medical conditions [45].

Besides the direct viral effect on muscle cells, the cytokine storm potentiated by this infection can support inflammatory effects at the muscular level. Thus, the growth of the C-reactive protein, marker of inflammation, the high levels of IL-1, IL-6 and TNF-a induce proteolysis at the level of muscle fibers, the activation of fibroblasts and the generation of fibrosis as well as the blocking of the progenitor cells responsible for the genesis of new muscle fibers [46,47]. These phenomena explain why the physical and respiratory recovery is so difficult after an episode of COVID-19 infection.

When evaluating the 6-min gait test, the patients showed lower results compared to the group of healthy patients. Lau HM et al. identified a reduction in the functional capacity of sick individuals, by 32% in terms of supporting the walking effort and by 13% in terms of the distance they managed to travel. This result brings to the attention the proinflammatory viral effects at the cellular level as a consequence the appearance of a deficit both at the level of muscular functional capacity and at the level of resistance to small efforts. Up to 36% of the patients included in the study required neurological consultations for the appearance of motor deficits. Their recovery period was long-term, so that 40% of the patients with the moderate/severe form of disease included in this study required recovery programs after COVID-19, returning to work only after 2–3 months after healing [12].

Muscle cell distructions due to proinflammatory reactions were suggested by Lee et al., who observed that hospitalized patients with moderately severe lung damage had creatine kinase (CK) levels between 269 and 609 U/L [48].

### 4.3. Sarcopenia

Acute sarcopenia was observed as a complication among patients infected with SARS-CoV-2. In an observational paper, Piotrowicz et al. talked about the mechanisms of pathophysiology and the management of post-COVID-19 sarcopenia. This complication can negatively influence the prognosis of patients, leading to physical and functional deterioration. The factors that could influence the development of sarcopenia were taken into account and it was noted that the state of health before infection, the treatment of severe inflammation given by the cytokine storm in the SARS-CoV-2 infection, as well as the avoidance of prolonged immobilization and adequate caloric intake can change the prognosis of the disease [24].

Sarcopenia is a reason for the impaired immunity because of the presence of IL-6, IL-15, IL-17, myokines that are responsible for the proliferation and function of the immune system. More than that, people with sarcopenia develop a metabolic stress, mainly defined by an excessive catabolism of the skeletal muscles for providing enough amino acids and glutamine for the whole organism [49].

The risk of acute sarcopenia and malnutrition has been shown to be higher among elderly patients, as claimed Morley et al. in their paper [25].

### 4.4. Muscle Catabolism

It has been proven that the functional capacity of COVID-19 patients is significantly reduced due to a metabolic dysfunction represented by muscle catabolism, which causes, as a consequence, changes in muscle fibers with a reduction in volume and mass [27,28].

Bed immobilization in any situation results in weakness and muscle damage. Kortebein et al. talks about the effects of a forced immobilization within 10 days and describes its effects. People with average ages between 62 and 72 years, immobilized for 10 days, showed a 6.3% reduction in muscle mass in the lower limbs, a decrease of up to 15.6% in the isokinetic force and up to 14% in the climbing force of the steps [50,51].

Mayer et al. reports an average decrease of 18.5% in femoris rectus muscles between the first day and the 7th day of admission of positive COVID-19 patients admitted to intensive care units [29].

Muscle cell destruction due to proinflammatory reactions was suggested by Lee et al., who observed that hospitalized patients with moderately severe lung damage had creatine kinase (CK) levels between 269 and 609 U/L [48].

### 4.5. Corticoids as a Therapy

Although the therapeutic management within this infection is already known, some therapies may interact with skeletal muscles. Corticoids, used in patients with respiratory failure, are responsible for the processes of muscle catabolism and lead to the loss of muscle mass [52,53]. Tocilizumab, an antibody against IL-6, used to reduce the cytokine storm, seems to increase anabolic muscle processes in the long-term, as demonstrated by DXA(Osteodensitometry) measurements in rheumatoid arthritis patients [30,54]. However, the effects of short-term therapy performed in COVID-19 patients are being researched. In addition to the effect of bed immobilization, muscle metabolism was also influenced by the pharmacological treatment administered to these categories of patients. The main therapeutic classes used in the treatment of COVID-19 infection were immunosuppressants, immuno-modular agents and corticotherapy. In addition to these, the patients benefited from ventilatory, hemodynamic and renal support. Although all these are absolutely necessary in the treatment of the disease, Schakman et al. recalls in a paper the fact that some of them have interactions with protein metabolism, causing a reduction in protein synthesis and degradation of muscle metabolism, contributing to the alteration of physical function, the accentuation of muscle weakness and a delayed recovery process [55].

In patients admitted to the intensive care unit, ventilatory, renal support, and hemodynamic stability are monitored under a regimen that includes high-flow nasal cannula, invasive mechanical ventilation, vasoactive therapy, and continuous renal replacement therapy. Currently, norepinephrine is recommended as the first choice, followed by vasopressin. Some of these drugs interfere with protein synthesis and muscle metabolism, which leads to impaired physical function in patients after COVID-19 infection. An example is systemic glucocorticosteroids that have well-known anti-inflammatory and immunosuppressive properties. Systemic glucocorticosteroid therapy is applied in clinical practice to inhibit the cytokine storm and to manage inflammation-induced lung damage in patients with COVID-19. Different treatment regimens are used depending on the patient’s characteristics and clinical severity [33].

During the 2003 SARS-CoV-2 pandemic, it was observed that the use of systemic corticosteroids to treat acute lung damage contributed to muscle atrophy and decreased functional capacity in patients. Corticosteroid therapy can also cause muscle damage by impairing the electrical excitability of muscle fibers, decreasing the number of thick filaments, and reducing anabolic protein synthesis along with increased protein degradation. Indeed, muscle atrophy is a well-known side effect observed in patients receiving long-term glucocorticosteroid therapy. Short-term administration of glucocorticosteroids has also been shown to induce early-onset myopathy in critically ill patients, which is characterized by a progressive weakening of several muscle groups. It should be noted that certain pharmacological interventions used during hospitalization in the intensive care unit may further exacerbate muscle damage [33].

Computer tomography is an essential method of diagnostic in COVID-19 disease and besides the important information that it helps us find, there are a lot more usage for this investigation. Yong You et al. described that CT is also used for diaphragm tickness parametres, which are related to the nutritional status and the guidance of nutritional is very important for the disease’s evolution. The diaphragm is the respiratory muscle that produces more than 60–80% of tidal volume in normal breathing. The prognosis of patients with COVID-19 pneumonia is influenced by the diaphragm atrophy and the authors suggested that monitoring these changes by using CT scan would have a clinical significance for the nutritional therapy and status [56].

Diaphragm thickness measurements obtained by using CT scan will prevent a high catabolism and muscle atrophy if the primary disease is recognised and treated and it can reflect the patient’s nutritional status. Diaphragm atrophy is associated with high catabolism in critical ill such as sepsis, systemic inflammation and trauma and CT-DT obtained can be an dynamic assessment tool with rapid results in the fight with the COVID-19 epidemic [34].

Mao L. et al., who conducted a study in China on a group of 214 hospitalized patients, pointed out that documented lung damage on CT imaging is a negative predictor factor for a patient’s respiratory and subsequent muscle recovery [3].

### 4.6. Muscle Fatigue

Muscle weakness was observed at patients with COVID-19, especially at the intesive-care admitted patients, with a 30% function reduction of the rectus femoris and 20% decrease in thickness of quadriceps muscles, after less than 10 days [26,35].

The strength of actions like knee extension and arm flexion were reduced in 75–85% of individuals from a cohort study, including hospitalized COVID-19 patients with ages between 40–88 years old. Muscle disabilities were correlated with a longer stay in the hospital and the recovery was complete only after 5 years, due to the associations of COVID-19 disease and ARDS (Acute respiratory distress syndrome) [19,36].

People with previous diagnosis of sarcopenia required a longer rehabilitation process and their rate of mortality was eight times higher than those without such a comorbidity. Other important risks for muscle alteration are age, being females, long stays at intensive-care, infections, sepsis and systemic inflammation [37].

Tissue impairments were suggested by the high echogenicity of the femoris muscle in patients with severe forms of COVID-19, possible due to the mentioned modifications in muscle tissue. Muscle atrophy is caused by the activation of enzyme ubiquitin proteasome which also causes the muscle autophagy, which brings the muscle loss in COVID-19 and other consumptive diseases like cancer, sepsis, cachexia and COPD (Chronic obstructive pulmonary disease) [38].

A cohort study conducted by Huang et al. on a group of 1733 patients admitted between 7 January 2020 and 29 May 2020 in Jin Yin-tan Hospital noted 63% of patients described fatigue or muscle weakness as the main symptomatology post-illness, 26% had sleep disorders and 23% were left with sequelae such as anxiety or depression [15].

Medrinal et al. conducted a study on a small group of 23 patients who required mechanical ventilation, discharged from intensive care units. A total of 30 days after the extubation, 69% of the patients had muscle weakness of the limbs and 26% showed respiratory muscle weakness. A total of 44% of the patients included in the study could not cover a distance of 100 m at the walking test [16].

### 4.7. Respiratory Rehabilitation, a Possible Cure?

The consequences of infection with SARS-CoV-2 virus are already known all over the world. Cured patients can be left with sequelae such as shortness of breath, fatigue, anxious syndrome and sometimes depression. This raised the question of the need for a pulmonary rehabilitation program, including respiratory exercises, education and psychological counseling, in order to improve the post-illness recovery process and increase the quality of the cured patients. Langer et al. described the benefits of pulmonary rehabilitation for patients with COPD after an 8-week home breathing exercise program. They have demonstrated the usefulness of such a training to reduce dyspnea, strengthen the inspiratory muscles and increase the capacity of physical effort [17].

Zamponga et al. conducted a study that followed the evolution of 140 patients integrated into pulmonary rehabilitation programs. Control tests were used before and after the completion of these recovery programs and the results were compared. The study tracked the capacity of the motor apparatus, monitored by barthel index (BI) with a score between 0–100, the muscles and the force of the lower limbs evaluated by tracking the short physical performance battery score (SPPB) and the ability to tolerate physical exertion using the 6 m walk test (6MWT). Following the introduction of pulmonary rehabilitation programs, led by a multidisciplinary team, with trainings whose intensity gradually increased, the patients presented the following results: an increase in the SPPB score from 4.2% to 66.7%; Bi increased from a score of 55 to one of 95. Regarding the walk test, when included in the program only 30% of the subjects managed to this test, then, after the completion of the program, 57.8% performed the test without problems [18].

An observational study on patients with severe forms of COVID-19 infection who required hospitalization in intensive care units was conducted by Van Aerde et al. The result of this study was to analyze the muscle weakness evaluated using the MRC-SUM (Medical Research Council sum) score. Of the 486 patients hospitalized in Leuven Hospital, between March and June 2020, 114 were transferred to the ICU wards and 74 were intubated. Fatigue and muscle capacity was evaluated at extubation, during hospitalization and discharge with the following results: 72%, 52% and 27%. This study wanted to bring to light the need to initiate post-COVID-19 recovery programs post-ICU, in order to speed up their healing and reduce the long-term impact of this disease [19].

The functionality of the main muscle groups was severely affected by the COVID-19 infection. A study conducted on muscle recovery after COVID-19, conducted by Paneroni et al., highlighted the reduction of muscle strength of the biceps and quadriceps muscle by up to 69% and 54% of the normal value, respectively, among 73% and 86% of patients taken into account [26].

Lau HMC et al. performed a study on a group of 133 patients with COVID-19 infection, and showed that those included in the pulmonary rehabilitation programs and respiratory physical therapy for a period of 6 weeks showed a 10% increase in lung function, improvement of effort adherence by over 17% and a significant increase in resistance to sustained efforts of flexion/extension of the arms and lower limbs. These programs, based on daily aerobic exercise, reduced the fatigue described by the patients and strengthened the respiratory, skeletal, bone and joint muscles, contributing to the increase of cardiopulmonary health [57].

Although initially the correlation between lung damage and severe myalguies suffered by patients diagnosed with COVID-19 was not considered relevant, later, Zhang et al. observed the association of the two, so that individuals who had a moderate/severe lung damage on CT also showed muscle pain and generalized weakness [58].

### 4.8. Early Mobilization for COVID-19 Patients

Valenzuela et al. describe in a meta-analysis that the early initiation of mobilization programs initiated since the period of hospitalization in COVID-19 patients hastens their healing and recovery. Following prolonged bed rest, patients acquire a hospital disability, which can have consequences even in the long term. Studies have reported that a 50-min daily exercise program, executed for 8 days, led to the reduction of dyspnea, the increase of muscle strength and functional capacity, thus leading to the improvement of the quality of life after discharge. Even for patients admitted to ICU wards, studies show that minimal stretching and lifting exercises led to a reduction in ventilator days, hastened the healing process, leading to a decrease in the number of days of hospitalization. The intubated patients, unable to mobilize, were given neuromuscular electrical stimulation sessions, through which the muscle contraction was achieved by applying electrical stimuli. This approach has alleviated the muscular melting of patients with serious conditions [20].

The factors that lead to the functional and muscular deterioration of the patients infected with SARS-CoV-2 have to do with both the degree of lesions they develop, the treatments they follow, as well as with the hospitalization itself. First of all, during the healing period of this disease, a prolonged rest is recommended to reduce metabolic consumption and conserve the body’s energy for recovery. However, this recommendation also has some negative sides. Bloomfield describes that the prolonged immobilization, over 3 weeks, frequently encountered among patients hospitalized in intensive care units, led to the alteration of the production of contractile proteins, with the melting of muscle mass and implicitly the decrease of muscle strength [59].

Ofori Asenso et al. talks about this negative impact of prolonged bed rest, which has proven to be more evident among the elderly, due to the aging process and the delayed sarcopenic phenomena and the significant loss of muscle mass [14].

Hospitalization rates for COVID-19 increase with age and the elderly have the highest risk of hospitalization. The negative functional consequences of prolonged hospitalization have been known for many years. In the context of COVID-19, epidemiological studies reported an average length of hospital stay of 20 days, with an average stay in the intensive care unit of three weeks [39]. Due to severe acute respiratory failure and the development of acute respiratory distress syndrome, many patients infected with COVID-19 require hospitalization in the intensive care unit. The interactions between critical complications related to the disease, comorbidity, treatments, organizational aspects of intensive care and adaptation in the post-therapy period, all these can contribute to the development of post-intensive syndrome. This syndrome is characterized by physical, functional, cognitive and mental changes and the development of post-traumatic stress, which can lead to decreased quality of life [40].

Associated comorbidities, such as old age, renal dysfunction, hypertension, diabetes, heart problems, multiple organ failure, all these factors contribute to the patient’s immobility, which in turn has detrimental effects on the cardiorespiratory system, central nervous system, muscle [40]. Evidence has shown that multiple organ failure is strongly associated with muscle dysfunction [60].

In infected patients, bed rest is recommended to minimize metabolic demand and to direct resources toward the recovery process. However, long periods of immobilization and hospitalization have been shown to have a negative impact on many body systems. Studies show that a period of 4 to 6 weeks of bed rest leads to muscle atrophy, loss of muscle strength from 6% to 40% of baseline muscle strength.In infected patients, bed rest is recommended to minimize metabolic demand and to direct resources toward the recovery process. However, long periods of immobilization and hospitalization have been shown to have a negative impact on many body systems. Studies show that a period of 4 to 6 weeks of bed rest leads to muscle atrophy, loss of muscle strength from 6% to 40% of baseline muscle strength [33].

Muscle fatigue or atrophy, difficulty sleeping, and anxiety and depression have been found to be common symptoms, even 6 months after COVID-19 infection. This is consistent with data from previous SARS long-term follow-up studies [15].

### 4.9. Myopathy and Polyneuropathy, a Possible Complication?

Knowing the frequency of these muscle complications in coronavirus disease 2019, studies have been conducted to seek the link of neuromuscular involvement and the SARS-CoV-2 virus. Thus, in a study on a group of 12 patients suspected of myopathy and polyneuropathy induced by the COVID-19 disease, Martinez et al. searched whether these two entities differ in characteristics from those induced by other pathologies. Nerve conduction tests and electromyography were performed in all 12 patients and muscle biopsy in three of them. Although only one such case has been previously reported, the authors want to emphasize that these pathologies will become more frequent in the future, and the correct evaluation and diagnosis, through nerve conduction studies and electromyography, of suspicious patients is absolutely necessary [21].

A 58-year-old woman was reported to have COVID-19 disease associated with myositis and bulbar weakness. Her blood tests showed CK level higher than 700 U/L. A magnetic resonance imaging test was made and reflected muscle edema with parts of myonecrosis. At the muscle biopsy they observed inflammatory infiltration, regenereating fibers and necrosis. Based on the presumptive diagnosis she received 5 days of 1 g intravenous methylprednisolone and Tocilizumab, based on high level of interleukin-6. After 2 weeks, the level of CK normalized and she started her recovery. She was able to move her arms and legs and her speech was improved. Viral infections are very well known for causing myositis. Zhang et al. reported this case with COVID-19 and myositis and invited other clinicians who had similar cases to add documentation to elucidate the mechanisms and adequate treatments for this muscular complications [23].

### 4.10. Psychologycal Disorders

PICS syndrome (post-intensive care syndrome) includes not only muscle destruction, and marked fatiguability, but also mental disorders, thinking, anxiety, depression and insomnia [31,61].

The current pandemic has significantly reduced people’s physical activity. The lock-down period had as a consequence the increase of sedentariness among the population. Using a multi-national web questionnaire conducted by 1047 people, Ammar et al. concluded that the level of physical activity decreased by 1/3 compared to the baseline level of 30 min of movement per day. Also, the time of stationary activities increased from 5 to 8 h/day [32].

## 5. Conclusions

Although over 2 years have passed since the onset of the pandemic due to the infection with the SARS-CoV-2 virus, the complex mechanisms by which the multisystem damage occurs are not fully known.

From the articles included we concluded that the musculoskeletal damage is firstly produced by the inflammatory effects, cytokine storm and muscle catabolism. However, myopathy, polyneuropathy and therapies such as corticoids were also considered important factors in muscle fatigue and functional impairement. Pulmonary rehabilitation programs and early mobilization had a high contribution during the acute phase and post-illness recovery process and helped patients to reduce dyspnea, increase the capacity of physical effort, overcome psychological disorders and improve the quality of their life.

## Figures and Tables

**Figure 1 medicina-58-01199-f001:**
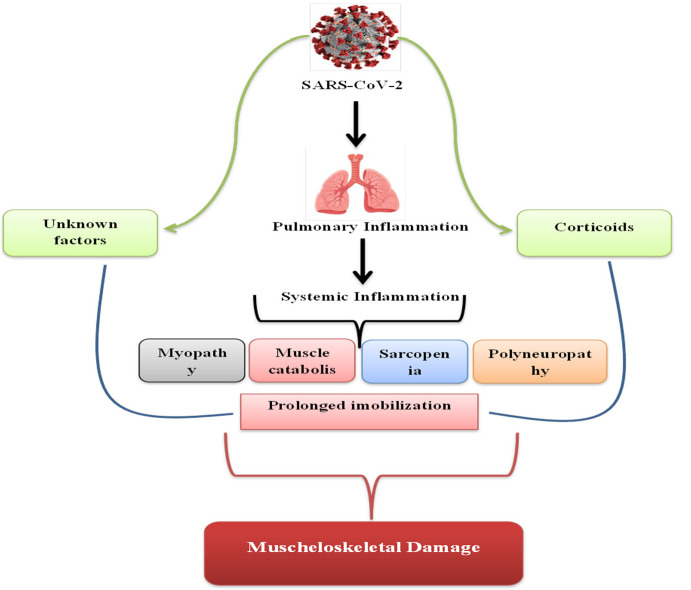
Pathogenic components leading to musculoskeletal injury in severe cases of COVID-19.

**Table 1 medicina-58-01199-t001:** Literature review of researched articles.

Authors	Type of Study	Pathology	Results
Disser NP et al., 2020 [11]	Prospective	COVID-19	Muscle, cortical and synovial tissues are direct locus of SARS-CoV-2 infection.
Lau HMC et al., 2021 [12]	Randomized Controlled Trial	Severe Acute Respiratory Syndrome And COVID-19	The pulmonary rehabilitation programs reduced the fatigue and increased the cardiopulmonary health.
Zhang et al., 2020 [13]	Cohort and Case Control	COVID-19	Moderate/severe lung damage on CT is linked to muscle pain and general weakness.
Mao L. et al., 2019 [3]	Observational Study	COVID-19	Lung damage CT imaging is a negative predictor factor for the patient’s recovery.
Ofori-Asenso R et al., 2019 [14]	Systematic Review and Meta-analysis	Frailty	The negative impact of prolonged bed rest is more evident among the elderly.
Huang et al., 2021 [15]	Cohort Study	COVID-19	The main symptomatology post-illness is fatigue or muscle weakness.
Medrinal et al., 2021 [16]	Observational Study	COVID-19	One month after extubation, the main symptoms were muscle weakness of the limbs and respiratory muscle weakness.
Langer et al., 2018 [17]	Clinical Trial	Chronic Obstructive Pulmonary Disease (COPD)	A 2-month respiratory exercise program reduced the dyspnea, strengthened the inspiratory muscles and increased the capacity of physical effort.
Zamponga et al., 2021 [18]	Retrospective Data Analysis	COVID-19	Pulmonary rehabilitation is possible and effective in patients recovering from COVID-19.
Van Aerde et al., 2020 [19]	Observational Study	COVID-19	Post-COVID-19 recovery programs post-ICU speed up the healing process and reduce the long-term impact of the disease.
Valenzuela et al., 2020 [20]	Meta-analysis	COVID-19	Early initiation of mobilization programs initiated since the period of hospitalization in COVID-19 patients hastens their healing and recovery.
Martinez et al., 2020 [21]	Retrospective Study	COVID-19	The myopathy and polyneuropathy induced by the COVID-19 disease does not have any distinctive features compared to those induced by other pathologies.
Tankisi H et al., 2020 [22]	Review	Critical Illness Neuropathy	The correct evaluation and diagnosis, through nerve conduction studies and electromyography is crucial in pacients suspicious of CIN.
Zhang et al., 2020 [23]	Case Reports	COVID-19-associated myositis	There’s a link between COVID-19 disease and inflammatory mediated myositis.
Piotrowicz et al., 2021 [24]	Review	Post-COVID-19 acute sarcopenia	The state of health before infection, the treatment, early mobilization and adequate caloric intake can change the prognosis of the acute sarcopenia COVID-19 induced.
Morley et al., 2020 [25]	Editorial	COVID-19	Elderly patients are at higher risk of acute sarcopenia and malnutrition.
Paneroni et al., 2021 [26]	Cross-sectional Study	COVID-19	The strength of the main muscle groups was significantly affected by the COVID-19 infection.
Kirwan R et al., 2020 [27]	Review	COVID-19, Sarcopenia	Physical activity, online and phone-based virtual care and telehealth services can protect vulnerable populations and maintain or improve the health status of the population at large.
Martinez-Ferran M. et al., 2020 [28]	Review	COVID-19-related Metabolic Dysfunctions	Adequate control of metabolic disorders could be important to reduce the risk of severe COVID-19.
Mayer KP et al., 2020 [29]	Observational Study	ICU-assessed Muscle Alterations	PPARβ/δ regulates FOXO1 activation in glucocorticoid- and sepsis-induced muscle wasting and that treatment with a PPARβ/δ inhibitor may ameliorate loss of muscle mass in these conditions.
Toussirot E et al., 2020 [30]	Multicenter Study	Rheumatoid Arthritis (RA)	Tocilizumab may have an anabolic impact on lean mass/skeletal muscle.
Michel J-P et al., 2020 [20]	Editorial	COVID-19	
Stam HJ et al., 2020 [31]	Cohort Study	COVID-19 and Post Intensive Care Syndrome	Post Intensive Care Syndrome and other severe conditions will require not only adequate screening but early rehabilitation and other interventions.
Ammar et al., 2020 [32]	Electronic Survey	ECLB-COVID19	While isolation is a necessary measure to protect public health, results indicate that it alters physical activity and eating behaviours in a health compromising direction.
Romero-Sagarra L., 2020 [33]	Review	COVID-19 and Muscular Weakness	Older people diagnosed with a COVID-19 infection are at a higher risk of severe muscle weakness and atrophy that may have negative consequences on functional disability.
Yong You et al., 2022 [34]	Retrospective Study	COVID-19 and Diaphragm thickness	CT-obtained DT can be used as a dynamic assessment tool for evaluating the nutritional status of patients in isolation wards for COVID-19.
Mario Chueire A. et al., 2021 [35]	Prospective study	Skeletal Muscle Wasting and Function Impairment in Intensive Care COVID-19 patients	In intensive care patients with severe COVID-19, muscle wasting and decreased muscle strength occurred early and rapidly during 10 days of ICU stay with improved mobility and respiratory functions, although they remained below normal levels.
Kim Ji-Won et al., 2021 [36]	Observational study	COVID-19 and Prognostic Sarcopenia for Length of Hospital Stay	Baseline sarcopenia was independently associated with a prolonged hospital stay in patients with COVID-19. Sarcopenia could be a prognostic marker in COVID-19.
Tao Yang et al., 2018 [37]	Meta-analysis study	COVID-19 and Risk factors for intensive care unit-acquired weakness	Acute Physiology and Chronic Health Evaluation II score, neuromuscular blocking agents and aminoglycosides were found to be significantly associated with ICUAW.
Sartori et al., 2021 [38]	Review	Mechanisms of muscle atrophy and hypertrophy	The lack of any efficient drug that counteracts muscle loss suggests that our view of the mechanistic insights that control atrophying muscle is still limited and needs further exploration.
Chaolin Huang et al., 2021 [15]	Cohort study	6 months consequences of COVID-19	COVID-19 and Patients who were more severely ill during their hospital stay had more severe impaired pulmonary diffusion capacities and abnormal chest imaging manifestations, and are the main target population for intervention of long-term recovery.
Carlotte Kiekens et al., 2020 [39]	Article	Rehabilitation and respiratory management in the acute and early post-acute phase	The first experiences in the field of rehabilitation show clearly the need to prepare for the post acute phase for patients who experienced a severe degree of the disease.
Bonorino K. et al., 2020 [40]	Article	Early mobilization in the time of COVID-19	Early rehabilitation interventions in patients with COVID-19, especially those who develop with severe muscle dysfunction, fatigue and dyspnea, (13) be initiated during hospitalization and continue in specialized rehabilitation programs after discharge.

Abbreviation: CT—computer tomography, ICU—intensive care unit, CIN—cervical intraepithelial neoplasia, PPARβ/δ—peroxisome proliferator-activated receptor, FOXO1—forkhead box protein O1, DT—delta time, ICUAW—intensive care unit acquired weakness.

## Data Availability

Not applicable.

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
