# Peer review of "The Effects of COVID-19 on Skeletal Muscles, Muscle Fatigue and Rehabilitation Programs Outcomes"

_medicina, 2022, doi:10.3390/medicina58091199_

Round 1

Reviewer 1 Report

Your manuscript has benn reviewed favorably and is being returned to you with minor comments for revision(included below). We look forward to receiving a revised manuscript.

The whole world is in trouble with COVID-19 infection and many are interested in its effects, especially on the body and its subsequent rehabilitation. I think this review will be positive and informative for many of those involved.

The content is carefully organized, and there are few corrections, but perhaps "31" on p8 L233 should be deleted, and the sentence on L236 is missing content. Please merge it with the paragraph following it or double check the content.

Reviewer 2 Report

Thank you for submitting the manuscript. I read your paper with great interest. The long-term consequences of Covid are an important topic and still not fully clarified.In this context your paper fits with authority as it is well done and structured. However I ask you to implement the introduction.In fact, the muscular sequelae of Covid cannot be disconnected from the neuritropism of the virus. In this regard, I suggest some references that can help you in expanding the introduction:

doi: 10.7759/cureus.13887.

doi: 10.2147/JPR.S313978.

doi: 10.3390/neurolint13010010.

I hope I have been helpful with these comments. Kind Regards
